# A Simulation Independent Analysis of Single- and Multi-Component cw ESR Spectra

Aritro Sinha Roy [1,2] , Boris Dzikovski [2] , Dependu Dolui [3] , Olga Makhlynets [4] , Arnab Dutta [3] and Madhur Srivastava [1,2,*]

1   Department of Chemistry and Chemical Biology, Cornell University, Ithaca, NY 14853, USA; as836@cornell.edu
2   National Biomedical Resource for Advanced ESR Spectroscopy, Cornell University, Ithaca, NY 14853, USA; bd55@cornell.edu
3   Department of Chemistry, Indian Institute of Technology Bombay, Mumbai 400076, India; arnabdutta@chem.iitb.ac.in (A.D.)
4   Department of Chemistry, Syracuse University, Syracuse, NY 13244, USA; ovmakhly@syr.edu
*   Correspondence: ms2736@cornell.edu

**Abstract:** The accurate analysis of continuous-wave electron spin resonance (cw ESR) spectra of biological or organic free-radicals and paramagnetic metal complexes is key to understanding their structure–function relationships and electrochemical properties. The current methods of analysis based on simulations often fail to extract the spectral information accurately. In addition, such analyses are highly sensitive to spectral resolution and artifacts, users' defined input parameters and spectral complexity. We introduce a simulation-independent spectral analysis approach that enables broader application of ESR. We use a wavelet packet transform-based method for extracting g values and hyperfine (A) constants directly from cw ESR spectra. We show that our method overcomes the challenges associated with simulation-based methods for analyzing poorly/partially resolved and unresolved spectra, which is common in most cases. The accuracy and consistency of the method are demonstrated on a series of experimental spectra of organic radicals and copper–nitrogen complexes. We showed that for a two-component system, the method identifies their individual spectral features even at a relative concentration of 5% for the minor component.

**Keywords:** ESR spectral analysis; hyperfine decoupling; resolution enhancement; wavelet packet transform; simulation-free spectra analysis

## 1. Introduction

ESR spectroscopy is a useful and powerful tool for studying biological free radicals and transition metal cofactors in proteins [1–7]. Understanding the electronic structure and local environment in such systems provides important insights into catalytic mechanisms [8–13] and redox processes in biological systems [6,14–16]. However, extracting information from ESR spectra can be challenging, especially in the case of weak and poorly resolved coupling interactions between studied spins. Traditionally, ESR spectral analysis is carried out by the spectral simulations, followed by user interpretation, which not only introduces bias but impairs the consistency of such analysis. The low-intensity signal components are difficult to analyze through standard spectral simulations in a variety of cases [17–19]. Accurate spectral analysis in such cases requires quantum chemical computations or special techniques [20–22], but without sufficient expertise and knowledge in the field, the interpretations rely heavily on the researchers' experience and intuition rather than the robustness of the method. An even more challenging problem, which occurs frequently, is the presence of more than one structurally similar molecule in a system. For closely resembling species, identifying multiple components by ESR alone could be a daunting task, and the standard simulation tools are incapable of extracting such information from the spectra without user

manipulation. Additionally, every standard ESR simulation method requires user defined starting configuration for optimization [23,24], which could lead to overfitting and/or unintended manipulation. Hence, the direct extraction of the relevant spin Hamiltonian parameters, namely the *g*-factors and and hyperfine coupling constant (*A*) values from an experimental ESR spectrum, would be ideal. The extracted parameter values can be used directly to interpret the electronic structure or, if needed, can be further optimized by standard spectral fitting software.

The capability of the wavelet transform to choose a periodic hyperfine pattern from poorly resolved ESR spectra was highlighted previously [25]. In a series of recent publications [26–28], we have presented an improved version of the wavelet transform based spectral analysis, which can decouple different frequency components in an ESR or nuclear magnetic resonance (NMR) spectrum, enhancing the resolution and providing an opportunity to extract spectral information or parameters in a selective and objective manner. Initially, we have modified the Noise Elimination and Reduction via Denoising (NERD) method [25,29], which is based on the discrete wavelet transform (DWT) method, for the separation of the hyperfine lines in cw ESR spectra and the extraction of spectral parameters [26]. Later, in analyzing $^1$H NMR spectra of molecular mixtures, which feature highly overlapped resonance lines due to the presence of scalar coupling between intramolecular protons, we recognized that spectral decomposition by the wavelet packet transform (WPT) is superior to DWT in separating the central frequencies from the multiplet structures encompassing them [27,28]. Using the same concept, a cw ESR spectrum is decomposed into its different frequency components by WPT, and at an optimum level of decomposition, a detail component in the wavelet domain is used to extract the hyperfine and/or superhyperfine structure, if any. The process is explained with an illustrative example later in this work (cf. Method). In the case of copper complexes, the analysis of the nitrogen-hyperfine structure is used for the determination of both $g_\perp$ and hyperfine splitting by copper, $A_\perp(Cu)$.

The outcome of the method has been validated by comparing the extracted hyperfine coupling constant values from both partially resolved and unresolved ESR spectra of Tempol and Tempo, recorded in the absence and in the presence of oxygen, respectively. We demonstrate the consistency of the WPT-based spectral analysis by using different wavelets in our analysis, namely Daubechies (Db6 and Db9) and Coiflet (coif3) wavelets. The method is applied across a wide range of copper complexes, and both the copper and nitrogen-hyperfine coupling constants are recovered from partially resolved and unresolved ESR spectra. The analysis did not use any prior knowledge about the structure of the complexes or user defined inputs in deriving the spectral parameters. The derived coordination geometry aligned with the structure predicted from analog studies, independent experiments and/or quantum computation, which further validates the results obtained by the WPT-based spectral analysis. The *g* and *A*-values obtained were compared with the optimized parameters obtained from spectral fitting by EasySpin software for a set of selected cases. The comparison shows that our method remains unaffected by artifacts, such as saturation and the passage effect, which affects simulations or spectral fitting significantly. While the performances of both methods were at par for well resolved spectra, unresolved spectral features remained inaccessible to the spectral fitting strategy. In those later cases, to the best of our knowledge, the proposed WPT-based analysis is the only method that can separate the so-called hidden features from poorly resolved spectra and extract the relevant spectral parameters by the direct analysis of an ESR spectrum. In this work, along with describing the extraction of spectral parameters by the WPT analysis of the ESR spectra of nitroxides and copper complexes, we demonstrated the efficiency of the method in identifying and analyzing multi-component spectra.

## 2. Method

### 2.1. Overview of the Wavelet Packet Transform Theory

A continuous wavelet transform can be defined as [30]

$$F(\tau, s) = \frac{1}{\sqrt{|s|}} \int_{-\infty}^{+\infty} f(\delta) \psi^* \left( \frac{\delta - \tau}{s} \right) dt \tag{1}$$

where $s$ is the inverse frequency (or frequency range) parameter, $\tau$ is the signal localization parameter, $\delta$ represents the chemical shift, $f(\delta)$ is the spectrum, $F(\tau, s)$ is the wavelet-transformed signal at a given signal localization and frequency, and $\psi^* \left( \frac{\delta - \tau}{s} \right)$ is the signal probing function called "wavelet". Different wavelets are used to vary the selectivity or sensitivity of adjacent frequencies with respect to signal localization. They are not dependent on *a priori* information of the signal or its characteristics.

Discrete wavelet transform (DWT) is expressed by two sets of wavelet components (detail and approximation) in the following way [30]:

$$D_j[n] = \sum_{m=0}^{p-1} f[\delta_m] 2^{\frac{j}{2}} \psi[2^j \delta_m - n] \tag{2}$$

$$A_j[n] = \sum_{m=0}^{p-1} f[\delta_m] 2^{\frac{j}{2}} \phi[2^j \delta_m - n] \tag{3}$$

where $f[\delta_m]$ is the discrete input spectrum, $p$ is the length of input signal $f[\delta_m]$, $D_j[n]$ and $A_j[n]$ are the detail and approximation components, respectively, at the $j$th decomposition level, and $\psi[2^j \delta_m - n]$ and $\phi[2^j \delta_m - n]$ are wavelet and scaling functions, respectively. The maximum number of decomposition levels that can be obtained is $N$, where $N = \log_2 p$, and $1 \leq j \leq N$. The scaling and wavelet functions, at a decomposition level, are orthogonal to each other, as they represent non-overlapping frequency information. Similarly, wavelet functions at different decomposition levels are orthogonal to each other.

The detail component $D_j[n]$ is the discrete form of Equation (1), where $j$ and $n$ are associated with $s$ and $\tau$, respectively. The approximation component $A_j[n]$ represents the remaining frequency bands not covered by the detail components until the $j$th level. The signal $f[\delta_m]$ can be reconstructed using the inverse discrete wavelet transform as follows:

$$f[\delta_m] = \sum_{k=0}^{p-1} A_{j_0}[k] \phi_{j_0,k}[\delta_m] + \sum_{j=1}^{j_0} \sum_{k=0}^{p-1} D_j[k] \psi_{j,k}[\delta_m] \tag{4}$$

where $j_0$ is the maximum decomposition level from which an input signal needs to be reconstructed. Compared to that, both the approximation and detail components at each level are further decomposed into a set of approximation and detail components. A schematic diagram of DWT and WPT decomposition against increasing levels are shown for comparison in Figure 1 [27].

### 2.2. A Case Study: WPT Analysis of cw ESR Spectrum of Tempo

In this section, we explain the details of WPT spectral analysis by using the cw ESR spectrum of Tempo as an example, shown in Figure 2. The ESR spectrum of Tempo is split into three major lines, corresponding to the hyperfine coupling of the $^{14}$N nucleus. Each of those lines are further split by the interaction of the $^1$H nuclei in the molecule, giving rise to what we designate as the superhyperfine splitting. Our goal is to separate the superhyperfine structure in the Tempo spectrum from the other spectral features. The first level of wavelet decomposition by the DB9 wavelet in Figure 2 shows a pair of approximate (A1) and detail (D1) components. It can be seen that D1 comprises noise, and the entire spectrum is represented by A1. Hence, D1 and all the components that derive from D1 during successive decomposition are not used in the spectral analysis. The decomposition

of A1 is continued until level-4, where the approximation component of AAA3 shows no superhyperfine splitting, and correspondingly, the approximation component derived from the decomposition of DAA3 contains the superhyperfine splitting pattern. This is why, in this case, decomposition until level-4 is considered to be optimal in separating the hyperfine and superfine splittings of Tempo's ESR spectrum. For cases where the superhyperfine is not resolved or visible, the optimum level of decomposition is chosen to be 4 based on previous work [26]. The WPT analysis is performed using Matlab software, version 9.12.0.1884302 (R2022a), and an illustrative code is given in Appendix A.1.

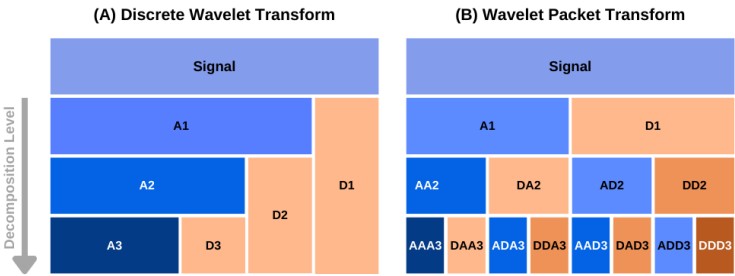

**Figure 1.** A schematic diagram of data decomposition in discrete (**A**) and packet wavelet transform (**B**) methods. The approximation and detail components at level $k$ are denoted as $A_k$ and $D_k$ in (**A**). In case of wavelet packet transform, the approximation and detail components at a decomposition level are denoted by the component name of the previous level followed by $A_k$ or $D_k$, respectively [27]. Copyright, 2022, The Journal of Physical Chemistry.

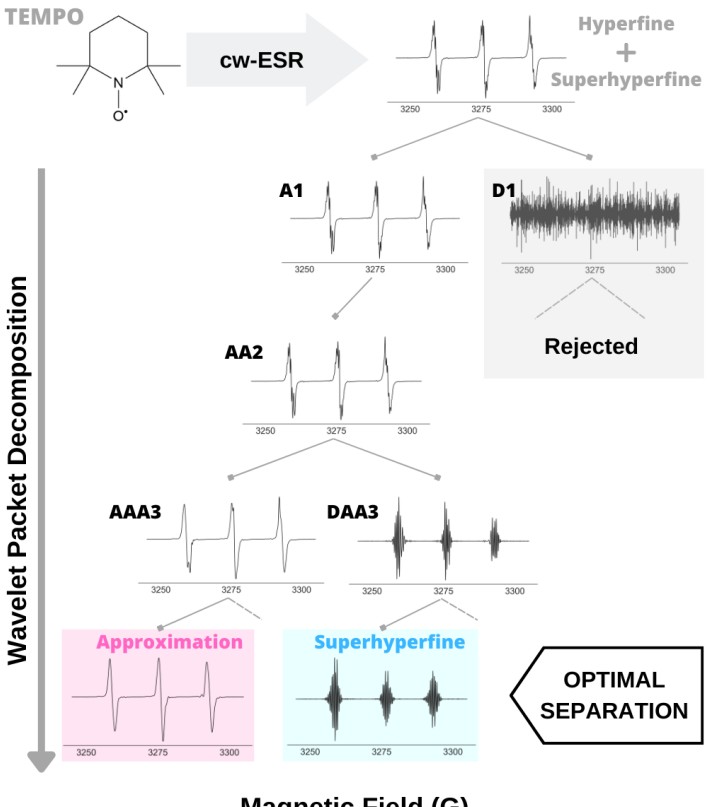

**Figure 2.** Separation of hyperfine and superhyperfine components in Tempo's ESR spectrum by WPT decomposition using Db9 wavelet. Given that at level-1, A1 contains all the spectral information, D1 and all the components derived from D1 are rejected. Complete separation of superhyperfine structure from the approximate component occurred after the decomposition level-3. Pure hyperfine (approximation) and superfine components are obtained from the decomposition of AAA3 and DAA3.

### 3. Materials

A series of experimental X-band ESR spectra were used in our analysis, and the corresponding molecular structures are given in Table 1. ESR spectra of Tempol and Tempo are very well studied, and we presented their spectral analysis by WPT for validation purposes. The mutation of superoxide dismutase-1 (SOD1) is arguably correlated with the incidence of significant fractions of familial and spontaneous cases of amyotrophic lateral schlerosis (ALS) disease [31,32]. For one of its mutants, SOD1:H48Q, the cw ESR spectrum was collected at 9.26 GHz, at a temperature of 30 K and concentration in the range of 300 to 350 μM using 50 μL 0.4 mm glass capillaries. No cryoprotectant was added given the viscous nature of the sample. Due to the presence of a glutamine residue in place of a histidine in the mutant, the nitrogen-hyperfine structure along the $g_\perp$ became partially resolved as a result of the reduction of the number of nitrogen in the copper coordination sphere. Cu-AHAHARA spectra were collected at 9.39 GHz for two different temperatures, 10 K and 100 K. ESR measurements of CuQu and CuQuA were performed in DMSO solutions at 100 K with an ESR frequency of 9.32 GHz.

**Table 1.** Molecular structures and ESR frequencies corresponding to the experimental cw ESR spectra used in this work.

| Molecule | Structure | ESR Frequency (GHz) |
|---|---|---|
| Tempo |  | 9.33 |
| Tempol |  | 9.33 |
| SOD1:H48Q | SOD1 mutant, histidine (48) replaced with glutamine | 9.26 |
| Cu-AHAHARA | A complex of Cu(II) and AHAHARA peptide: C-terminus as amide and N-terminus as acetyl group | 9.39 |
| CuQu |  | 9.316 |
| CuQuA |  | 9.316 |

*3.1. Experimental Section*

Synthesis

**Copper quinoline (CuQuA):** 160.0 mg (1.0 mM) 2-amino-8-quinolinol was taken in an 80.0 mL Schlenk flask with a magnetic stirrer bar, and 15.0 mL methanol was added to

make it a homogeneous yellow color solution. Then, 85.0 mg (0.49 mM) $CuCl_2 \cdot 2H_2O$ was taken separately with 10 mL MeOH in another Schlenk flask under steam of $N_2$. Then, the two methanolic solutions were charged under nitrogen flow. An immediate dark brown solution appeared after adding the metal salt to the ligand. Reducing the solvent with continuous $N_2$/Ar flashing, whitish deep brownish precipitate was observed. The reaction mixture was further stirred for two hours for completion with $N_2$ purging. Then, the precipitate was collected, washed with hexane and diethyl ether and dried under a high vacuum (yield = 137.0 mg, 76% w.r.t. 2-amino-8-quinolinol). XRD-suitable crystals (CCDC 2127156) were grown from slow diffusion from methanol/diethyl ether solution.

**Copper quinone (CuQu):** Synthesized as reported above, where 8-hydroxyquinoline was utilized as the precursor ligand (yield = 120.0 mg).

**Copper AHAHARA:** The AHAHARA peptide was synthesized by manual Fmoc solid-phase synthesis at elevated temperature using Amide Rink resin, Fmoc-protected amino acids and previously reported protocols [33].

**SOD1:H48Q:** The recombinant SOD1:H48Q protein, replacing a histidine with glutamine, was expressed and purified as described in a previous work [34].

### 3.2. ESR Experiments

Oxygen-free samples of Tempol and Tempo were prepared by flame-sealing after a triple repeat of the freeze–thaw cycle under vacuum. Oxygen-saturated samples were prepared by passing oxygen gas through the radical solution in water for 10 min and kept under oxygen atmosphere during measurements. Samples with an intermediate oxygen concentration were prepared and handled in air. The nitroxide concentration in all the samples was 100 μM. The ESR spectra were recorded at 293 K at a microwave (MW) frequency of 9.33 GHz, power of 0.1 mW, modulation frequency of 100 KHz and modulation amplitude of 0.1 G. The ESR spectrum of SOD1:H48Q was recorded at 30 K at an MW frequency of 9.26 GHz, power of 0.06325 mW, modulation frequency of 100 kHz and modulation amplitude of 4 G. The copper concentration was estimated to be 50 μM. The ESR spectra of Cu-AHAHARA complex were acquired in Wilmad tubes using a Bruker Elexsys E500 EPR spectrometer equipped with a cryostat. Peptide stocks at pH 2 were prepared fresh by adding lyophilized solid (>90%) to 10 mM HCl until peptide concentration reached 1 mM. First, Cu(II) solution in water (1 mM, 75 μL) and peptide stock (1 mM in 10 mM HCl, 150 μL) were mixed, and then buffer (91 mM Hepes, pH 8, 275 μL) was added to make a solution with 150 μM Cu(II) and 300 μM peptide (500 μL final volume). The mixture was incubated at room temperature for 2 h, and then glycerol was added to a final concentration of 10%, transferred into an ESR tube and flash frozen in liquid nitrogen. The final pH of the sample was 7.6 as measured by a Spintrode electrode (Hamilton). ESR spectra were acquired at 100 K and 10 K using the following conditions: frequency 9.39 GHz, power 5 mW or 2 mW, modulation frequency 100 kHz, modulation amplitude 8 G (10 K) and 4 G (100 K) and time constant 163.8 ms. The ESR spectra of CuQu and CuQuA were collected using the Bruker cw ESR EMX spectrometer at the National Biomedical Resource for Advanced ESR Spectroscopy (ACERT). About 100 mg of the air-stable copper complexes were dissolved in 1 mL DMSO. The solutions were poured in 4 mm ESR tubes, frozen in liquid nitrogen and kept in liquid nitrogen Dewar until they were inserted in the spectrometer. The spectra were recorded at 100 K, a microwave frequency of 9.316 GHz and attenuation of 30 dB for CuQu and 50 dB for CuQuA.

### 3.3. ESR Spectral Mix

For multi-component spectral analysis, four mixed spectra were calculated by mixing the ESR spectra of the compounds, CuQu and CuQuA, in proportions of (A) 2:1, (B) 4:1, (C) 10:1 and (D) 20:1 (Figure 3).

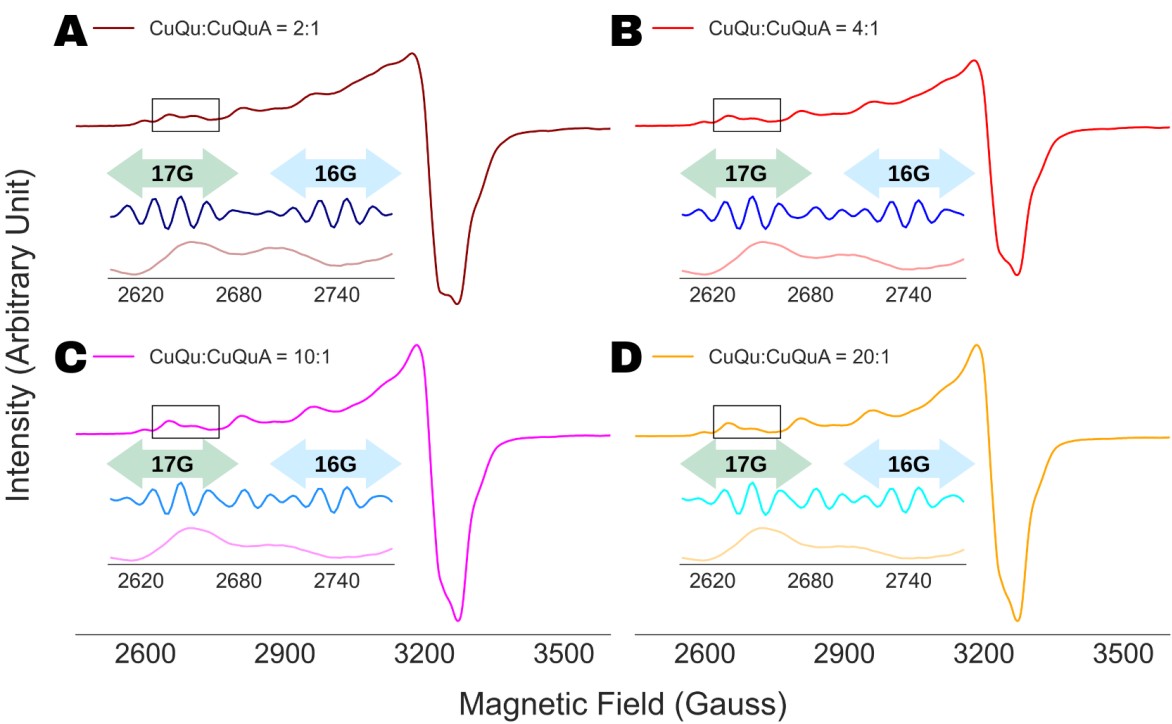

**Figure 3.** WPT analysis of ESR spectra of mixtures of CuQu and CuQuA, mixed in proportions of (**A**) 2:1, (**B**) 4:1, (**C**) 10:1 and (**D**) 20:1, respectively. The insets show the WPT component for the spectral region between 2600 G and 2775 G. Two components were identified in each of the spectral analyses from the difference in nitrogen-hyperfine splitting of 17 G (green double headed arrow) and 16 G (blue double headed arrow).

## 4. Results and Discussion

### 4.1. Validation of the Analysis

The interpretation of partially resolved ESR spectra by fitting has been the standard practice in the study of organic radicals and paramagnetic metal complexes. The approximations and the principles of such optimizations remain unknown to the users, whose interpretation of the results is largely dominated by the goodness of the fit and a priori knowledge about the structure or nature of the molecule under investigation. In contrast, a direct extraction of parameters, a process that lacks visual aids, can raise queries about the accuracy of the outputs. That was our motivation to run the analysis on a series of X-band ESR spectra of Tempol and Tempo under varying oxygen concentrations, as shown in Figure 4. The superhyperfine lines are fully or partially resolved in absence of oxygen or when it is present in very low concentration. However, the features become invisible due to line broadening by increased oxygen concentration. Consequently, while the superhyperfine constant in the former cases could be obtained by simple visual inspection or spectral fitting, none of those are applicable in the latter. In all those cases, the WPT-based method recovered the superhyperfine structure from the corresponding ESR spectra and extracted the value of the coupling constant, illustrating both its advantage over the existing spectral analysis approaches and its consistency.

It should be noted that the hyperfine components contain some features in between the regions of interest, clearly visible in Figure 4(B1). Such artifacts can be discarded by either (i) comparing with the original spectra (which was the case for Figure 4(B1)) or (ii) analyzing the inter-peak spacing, which yields non-uniform splitting in case of artifacts.

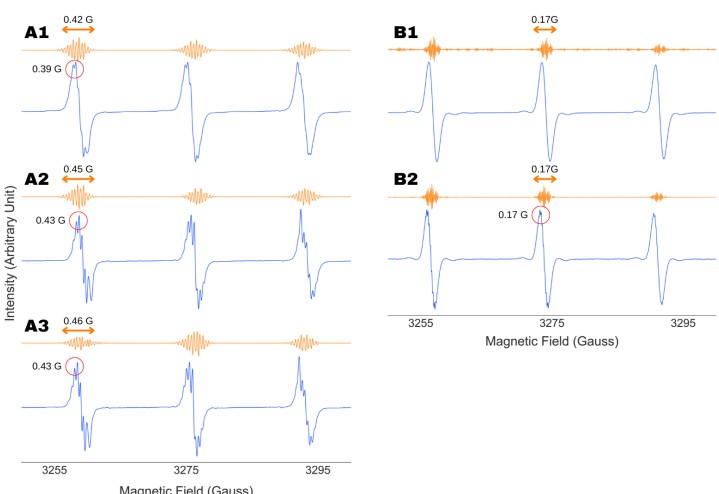

**Figure 4.** Analysis of X-band cw ESR spectra (blue) of Tempol (**A**) and Tempo (**B**) under varying concentrations of oxygen. The superhyperfine splitting due to the protons is partially resolved in (**A(1–3),B2**). The superfine splitting is invisible in the fully oxygenated Tempo spectrum (**B1**) due to line broadening. The superfine spectra recovered by the wavelet packet transform-based analysis of the spectra are shown (orange). The similarity between the superfine splitting constant obtained from the WPT analysis and direct analysis of the partially resolved should be noted. The validity of the analysis in case of an unresolved spectrum is demonstrated for (**B1,B2**).

### 4.2. Robustness Against Wavelet Selection

An important and non-trivial variable in the WPT analysis of ESR spectra is the selection of a wavelet, or in other words, understanding the robustness of the analysis against the types of wavelets used in the analysis. We repeated the analysis shown in Figure 4 with three different types of wavelets, Coiflet-3 (Coif3), Daubechies-6 (Db6) and Daubechies-9 (Db9), and the results are summarized in Figure 5. The extracted superhyperfine constant values for Tempol and Tempo, $0.44 \pm 0.01$ G and $0.20 \pm 0.01$ G, showed insignificant variation for all three wavelets while producing the same amount of splitting across all the analyses, consistent with the molecular structure. This observation illustrated the robustness of the method while further validating the accuracy of the extracted parameters. For the rest of the analysis in this work, we used the Db9 wavelet.

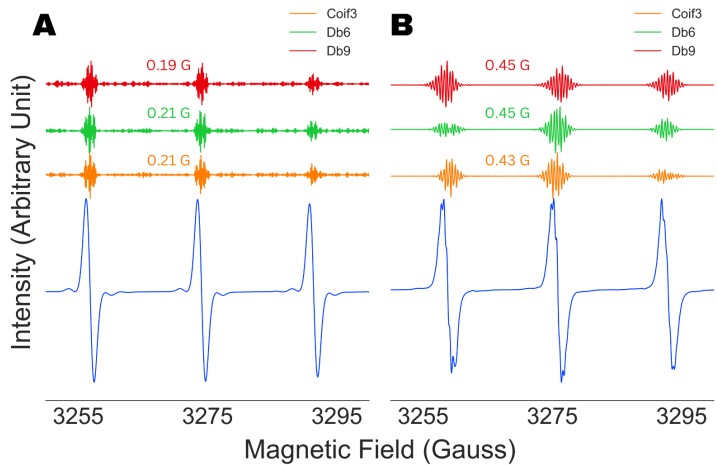

**Figure 5.** Analysis of unresolved and partially resolved X-band cw ESR spectra (blue) of Tempo (**A**) and Tempol (**B**). The superfine spectra recovered by the wavelet packet transform-based analysis of the spectra are shown using three different wavelets, Coiflet-3 (orange), Daubechies-6 (green) and Daubechies-9 (red). Superfine coupling constants obtained are consistent across the analysis involving three different wavelets.

### 4.3. Spectral Analysis of Partially Resolved ESR Spectra

We started our analysis by fitting the X-band cw ESR spectra of SOD1:H48Q recorded at 30 K and Cu-AHAHARA, recorded at 10 K and 100 K by using the EasySpin software. The optimized simulations along with the experimental spectra are shown in Figure 6. For SOD1:H48Q, the EasySpin analysis yielded nitrogen-hyperfine constant values of 15.6 G (axial) and 9.0 G (parallel), while the simulations for Cu-AHAHARA spectra did not require any nitrogen-hyperfine parameter in fitting the spectra. It should be noted that the Cu-AHAHARA spectrum at 10 K shows no nitrogen-hyperfine splitting; however, partial nitrogen-hyperfine splitting is evident for the spectrum at 100 K. This apparent anomaly might have resulted from the partial saturation and passage effect [35,36] in the case of the former, as evidenced by the behavior of the first integral of the spectrum, and consequently, the EasySpin fit for the 10 K spectrum performed poorly compared to that of the 100 K spectrum. In addition, the simulation presented for SOD1:H48Q in Figure 6A emphasized the potential presence of a second component, with a slightly shifted $g_\parallel$ and/or different hyperfine coupling constants.

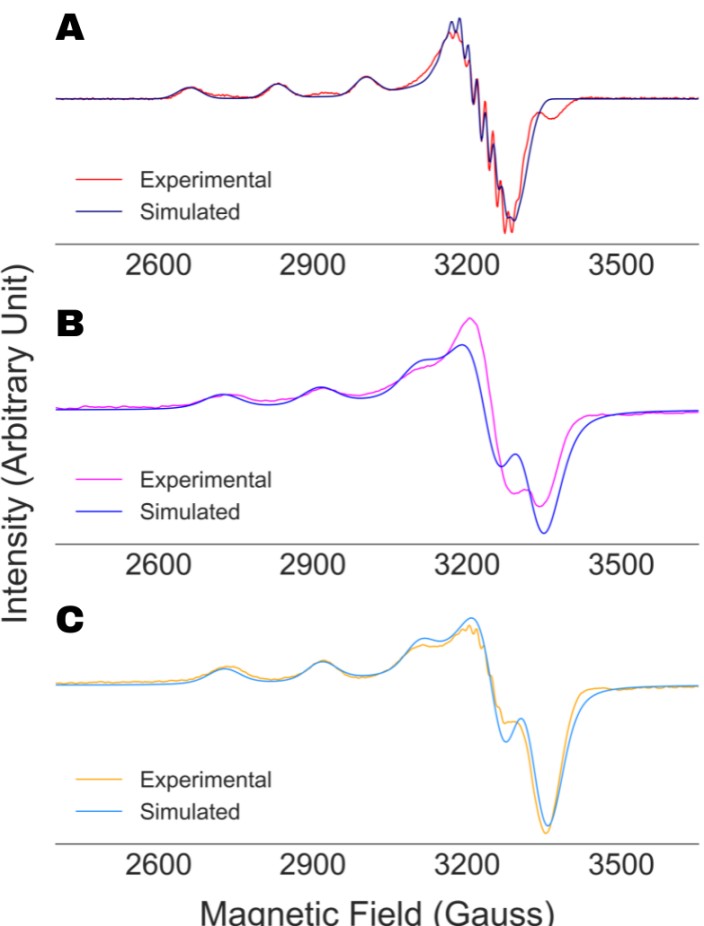

**Figure 6.** EasySpin simulation results along with experimental ESR spectra of SOD1:H48Q (**A**), Cu-AHAHARA spectrum recorded at 10 K (**B**) and 100 K (**C**).

In Figure 7, we present the WPT analysis of the ESR spectra of (A) SOD1:H48Q, (B) Cu-AHAHARA at 10 K and (C) Cu-AHAHARA at 100 K. For all the analyses, the spectra were decomposed to level-4 using the DB9 wavelet. For SOD1:H48Q, our analysis yielded a nitrogen-hyperfine coupling constant of 14.1 G for the axial peaks in the range of 3200 to 3320 G. However, an analysis of the $g_\parallel$ splitting (Figure 7(A.2)) revealed two different nitrogen-hyperfine splitting constants: 12.5 G along the dominant Cu parallel hyperfine splitting and 16.5 G along the minor component. Following that, we analyzed

the splitting along the small axial peak between 3220 and 3400 G (not shown), which emphasized the presence of four nitrogens and a nitrogen-hyperfine constant of $A_N$ = 16.5 G. From this analysis, we could infer a spherical electron density distribution in the second component, suggesting a tetrahedral copper complex with a large hyperfine splitting of ∼205 G. Similar analysis for Cu-AHAHARA yielded nitrogen-hyperfine coupling constants of 13.9 G (10 K) and 14.0 G (100 K) along the $g_\perp$ signal component. The consistency of the results demonstrates the robustness of the WPT-based spectral analysis against artifacts, such as an over-saturation effect in this case, which is a major advantage over the standard spectral fitting methods.

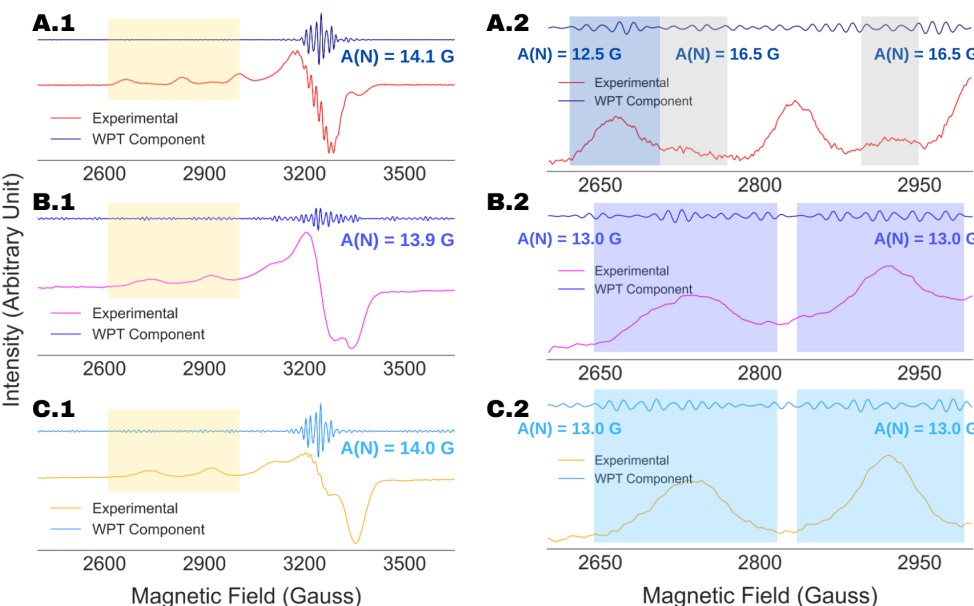

**Figure 7.** Recovery of nitrogen-hyperfine features by WPT analysis from experimental cw ESR spectra of SOD1:H48Q (**A**), Cu-AHAHARA at 10 K (**B**) and Cu-AHAHARA at 100 K (**C**). The left panel illustrates analysis of the $g_\perp$ component, while the $g_\parallel$ component is highlighted in yellow and the corresponding analysis is shown on the right. For SOD1:H48Q (**A**), resolving the nitrogen-hyperfine splitting along $g_\parallel$, a dominating component with $A_N$ = 12.5 G (shaded blue) and a minor component with $A_N$ = 16.5 G (shaded gray) were obtained.

It should be noted that the copper coordination geometry of the Cu-AHAHARA complex could not be confirmed in a previous work from the ESR spectral analysis [33]. The WPT analysis presented in Figure 7 suggests a strong overlap of nitrogen-hyperfine components in the $g_\perp$ region of the ESR spectra at 10 K and 100 K, while the former seems to contain spectral artifacts, making analysis by standard procedure error-prone. Hence, we analyzed the WPT components originating from the $g_\parallel$ region for the both the cases, which is summarized in Figure 8. In the case of the spectrum collected at 10 K, the WPT component originating from the ESR peak centered at 2730 G showed nine evenly spaced lines at 12.5 G apart from each other, indicating an $N_4$-coordination for the copper center in the Cu-AHAHARA complex. In addition, a smaller coupling constant in the $g_\parallel$ region in comparison to the value obtained in the $g_\perp$ region (13.9 G) suggests four equivalent nitrogens in the equatorial plane. This interpretation aligns well with a series of independent experimental and theoretical studies conducted to elucidate the previous geometry of the complex [33]. For the spectrum recorded at 100 K, a similar analysis in the $g_\parallel$ region did not reproduce the exact same results because of unresolved spectral overlapping in the WPT component. However, upon close inspection, we resolved half of the nitrogen-hyperfine splitting window for the $g_\parallel$ peaks centered at 2734 G and 2901 G. In this regard, it should be noted that only the even spacing between the peaks in a WPT component around a $g_\parallel$ peak was used as the criteria for recovering nitrogen-hyperfine splitting. The WPT analysis is highly accurate in recovering spectral information

with respect to their location but not necessarily the intensity. Factors such as the partial overlapping of resonance lines, residual noise and spectral artifacts affect the intensities of the peaks in a WPT component, and hence the recovered superfine splitting is unlikely to reproduce the relative intensity pattern expected for perfectly resolved spectra. The analysis of the Cu-AHAHARA ESR spectrum at 100 K yielded the same nitrogen-hyperfine coupling constant of 12.5 G and suggested an $N_4$ coordination for the copper center.

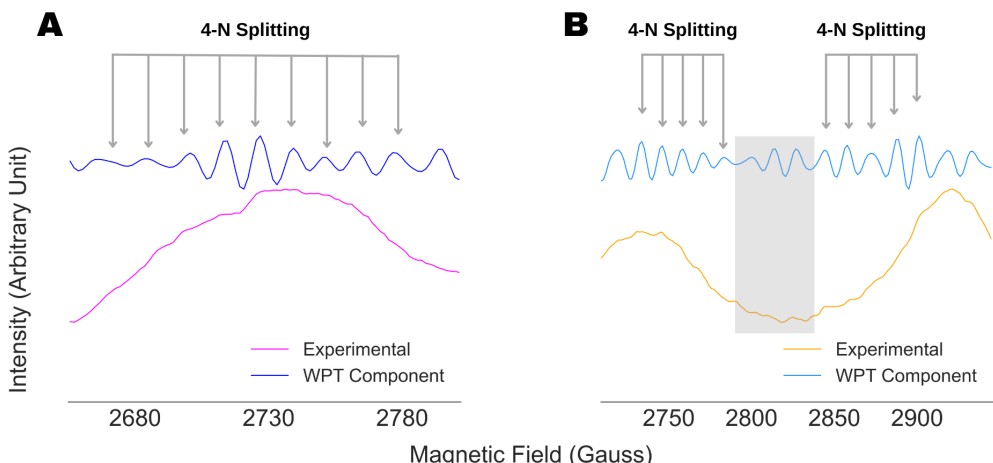

**Figure 8.** Analysis of the $g_\parallel$ components in the ESR spectra of Cu-AHAHARA complex at (**A**) 10 K and (**B**) 100 K for probing the copper coordination. For (**A**), the splitting pattern observed in the WPT component between 2650 G and 2800 G revealed nine equally spaced lines centered at 2727 G with an inter-peak spacing of 12.5 G, indicating four nitrogens in the copper coordination sphere. In the case of (**B**), the entire ranges of nitrogen-hyperfine splitting for none of the $g_\parallel$ components were visible due to overlapping in the WPT component. However, for both the $g_\parallel$ components centered at 2734 G and 2901 G, half of the nitrogen-hyperfine splitting windows were resolved, indicating four nitrogen coordination geometry for the copper center with a coupling constant of 12.5 G. The WPT component in the shaded region (gray) was not considered in the analysis because of varying inter-peak spacing in the region.

The WPT analysis for the X-band ESR spectra of two copper–nitrogen complexes, CuQu and CuQuA, are shown in Figure 9. The detail component in the optimal level of decomposition, which was three for both the spectra, revealed an unresolved hyperfine structure due to nitrogens coordinated to the copper centers in the complexes. For example, Cu(II) splits the resonance line at $g_\perp$ into four lines—we call them h-1, h-2, h-3 and h-4 for explanation purposes. Each of these lines split further due to the $n$ interacting nitrogens into $m = (2 \times n + 1)$ lines. Given the small magnetic field window between h-1 and h-4, as well as the nitrogen-hyperfine coupling, most of the resonance lines cannot be resolved due to complete or partial overlapping. However, for most of the cases, it might be possible to identify the first $m/2$ lines originating from h-1 and the last $(m+1)/2$ lines originating from h-4, where the extent of overlapping of resonance lines is the least. Using this logic, from the detail component analysis for CuQu in Figure 9A, we identified three lines with a total separation of 53 G, which corresponded to an $m = 5$ or two coupled nitrogens with h-1 and h-4 at 3203 G and 3314 G. Further, we calculated the nitrogen-hyperfine coupling constant to be $A(\mathrm{N}) = 53/2$ G or 26.5 G, the Cu(II) hyperfine coupling $A(\mathrm{Cu}) = (3314 - 3203)/3$ G or 37 G and the $g_\perp = (3314 + 3203)/2$ G $\equiv 2.0427$. The calculated spin-Hamiltonian parameters are in good agreement with previously reported analogous $N_2O_2$-coordinated copper(II) complexes [37,38]. With a similar analysis for CuQuA, Figure 9B, we inferred that three nitrogens were coordinating with the Cu(II) center with $A(\mathrm{N}) = 21.3$ G, while the $g_\perp$ and $A(\mathrm{Cu})$ were calculated to be 2.0411 and 23.3 G. The original $N_2O_2$ coordination geometry for the copper center observed in CuQu complex can be expanded in CuQuA due

to the presence of peripheral amine in the QuA ligand scaffold. The potential involvement of one amine group is supported by the EPR data, indicating an $N_3O_2$ coordination.

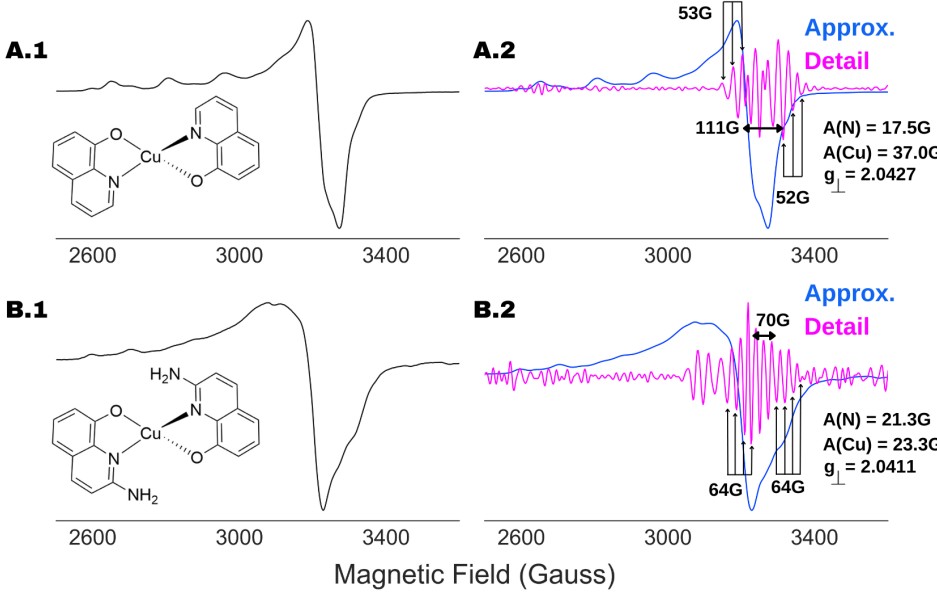

**Figure 9.** Recovery of nitrogen-hyperfine structure and coordination of Cu(II) center in (**A**) CuQu and (**B**) CuQuA. Both the ESR spectra (left panel) were recorded at 100 K and an ESR frequency of 9.316 GHz with attenuation of (**A**) 30 db and (**B**) 50 db, respectively. The approximation (blue) and detail (magenta) spectral components originated at the optimal decomposition level of 3 are shown in the right panel (scaled arbitrarily for visualization purpose). For both the cases, the nitrogen superfine structure were identified, showing (**A**) 2-N and (**B**) 3-N coordination to the copper centers in the two cases, along with the position of $g_\perp$ and hyperfine coupling due to Cu(II).

### 4.4. Analysis of Multi-Component ESR Spectra

It has already been shown in Figure 7 that the spectral analysis of the $g_\parallel$ components in the ESR spectrum of SOD1:H48Q revealed the presence of two components. In this section, we present further proof of how the WPT-based spectral analysis can be utilized efficiently to identify multi-component spectra, even when the components are present in highly disproportionate amounts. A set of four mixed spectra was produced for the analysis by mixing the ESR spectra of CuQu and CuQuA in proportions of (A) 2:1, (B) 4:1, (C) 10:1 and (D) 20:1, shown in Figure 3. The WPT spectral analysis was conducted using Db9 wavelet at decomposition level of 4, and the WPT detail components for the spectral region between 2600 and 2775 G are shown in the insets of Figure 3. In all the cases, five lines with a splitting constant of 17 G were recovered between 2610 and 2670 G, which corresponds to CuQu, which is expected because of the abundance of the complex in the mixtures. However, a second component with a splitting constant of 16 G was identified in between 2700 and 2748 G. For the latter, only four or three lines were visible due to spectral overlap, but it can be separated from artifacts by the consistent positions of the peaks appearing the detail WPT component. It should be noted that this strategy clearly identifies a second minor component directly from the spectral analysis, in this case CuQuA, but it may not be possible to determine the structure of the component solely from the analysis due to spectral overlaps, leading to unrecoverable information loss. However, we would like to emphasize that we used poorly resolved X-band ESR spectra of the individual complexes, and given such constraints, we believe that it is a major achievement to detect a second component solely from ESR spectral analysis, even when the second component is present at as low a relative concentration as 5%. In general, such findings from the WPT spectral analysis will help researchers to decide on further structural analysis by employing high-frequency ESR and/or complementary techniques.

## 5. Conclusions

In this work, we have provided the complete recipe for a simulation-free analysis of cw ESR spectra using a wavelet packet transform-based algorithm and experimental X-band ESR spectra of organic free radicals (Tempol and Tempo) and metal complexes (copper (II) inorganic and biological complexes). We showcased three major accomplishments of the WPT-based spectral analysis over standard simulations: (1) direct extraction of nitrogen-hyperfine structure from poorly resolved/unresolved spectra, (2) insensitivity toward spectral artifacts and (3) identification of multi-component samples. A consistent extraction of the superhyperfine coupling constant due to protons from both partially resolved and unresolved spectra for Tempol is presented, validating the accuracy of the new spectral analysis technique. The limitations of standard ESR spectral analysis are illustrated by running EasySpin simulations for partially resolved and poorly resolved spectra, with one of the cases containing some experimental artifacts. While the accuracy and efficiency of the simulations varied drastically in those cases, the WPT-based analysis extracted spectral parameters, revealing hyperfine and superhyperfine splittings not recoverable by EasySpin simulations, and the analysis was unperturbed by spectral artifacts. For the copper–nitrogen complexes, the resolved hyperfine structures confirmed the number of nitrogen atoms coordinating with copper centers as well as their coordination geometry. Finally, by close examination of the detail WPT components originating from the $g_{\parallel}$-regions of the spectra, the presence of two different components in the case of SOD1:H48Q, a mutant of SOD1, has been confirmed along with their nitrogen-hyperfine coupling constants. The strength of the method in resolving multi-component spectra has been displayed further by analyzing four mixed spectra of two copper–nitrogen complexes, CuQu and CuQuA. The spectral analysis revealed the presence of the two components even when the proportion of the two components in the spectrum was 20:1. In addition, three different wavelets were used for selected cases, emphasizing the robustness of the method against the choice of wavelets. The comprehensive analysis presented in this work is expected to motivate broad adoption of the technique in analyzing ESR spectra, and the spectral parameters obtained from the analysis can be further optimized using simulations and/or other computational methods.

**Author Contributions:** Conceptualization, A.S.R., B.D. and M.S.; methodology, A.S.R., B.D. and M.S.; formal analysis, A.S.R. and B.D.; material synthesis, A.D., B.D., D.D. and O.M.; spectral data collection, A.S.R., B.D. and O.M.; investigation, A.S.R.; writing—original draft preparation, A.S.R.; writing—review and editing, O.M., A.D. and M.S.; visualization, A.S.R.; supervision, M.S.; project administration, M.S.; funding acquisition, M.S. and A.D. All authors have read and agreed to the published version of the manuscript.

**Funding:** Data were collected at the ACERT facility for ESR on Cornell's campus (the National Institute of General Medical Sciences/National Institutes of Health under grant R24GM146107). This research was funded in part by the DST-SERB ore research grant (CRG/2020/001239).

**Institutional Review Board Statement:** Not applicable.

**Informed Consent Statement:** Not applicable.

**Data Availability Statement:** The data used in this paper can be accessed on 15 April 2023 via https://github.com/Signal-Science-Lab/Simulation_Independent_ESR_Spectral_Analysis.

**Conflicts of Interest:** The authors declare no conflict of interest.

## Abbreviations

The following abbreviations are used in this manuscript:

| | |
|---|---|
| ESR | Electron spin resonance |
| WPT | Wavelet packet transform |
| NERD | Noise Elimination and Reduction via Denoising |
| SOD1 | Superoxide dismutase-1 |

## Appendix A. Analysis of 100 K Cu-AHAHARA ESR Spectrum

*Appendix A.1. Matlab Code for WPT Decomposition Shown in Figure A1*

```
% set directory
cd "user directory";

% wavelet decomposition level
nlevs = [3,4,5];

% load data
fname = 'AHAHARA_100K.txt';
tag = strtok(fname, '.');
df = readtable(fname);

for i = 1:3
    % apply wavelet packet transform
    n = nlevs(i);
    uwpt = wpdec(data.Var2, n, 'db9');

    % approximation component
    rwpc0 = wprcoef(uwpt, 2^n - 1);
    % hyperfine component
    rwpc1 = wprcoef(uwpt, 2^n + 1);

    % save the results
    writetable(table(df.Var1, rwpc0),
                strjoin([tag,"_aL",int2str(n),".txt"],''));
    writetable(table(df.Var1, rwpc1),
                strjoin([tag,"_hL",int2str(n),".txt"],''));
end
```

*Appendix A.2. Spectral Analysis*

We plot the approximation and hyperfine components from the decomposition of Cu-AHAHARA ESR spectrum collected at 100 K in Figure A1, following the procedure described in the case study of Section 2.2. It can be seen that in successive wavelet decomposition of the spectrum using the Db9 wavelet, the approximation component at level-5 develops spectral distortion in comparison to the original spectrum. Therefore, we select level-4 as the optimal decomposition level and analyze the hyperfine component at that level to probe nitrogen hyperfine splitting. The next steps in the analysis and the results are given in Section 4.3.

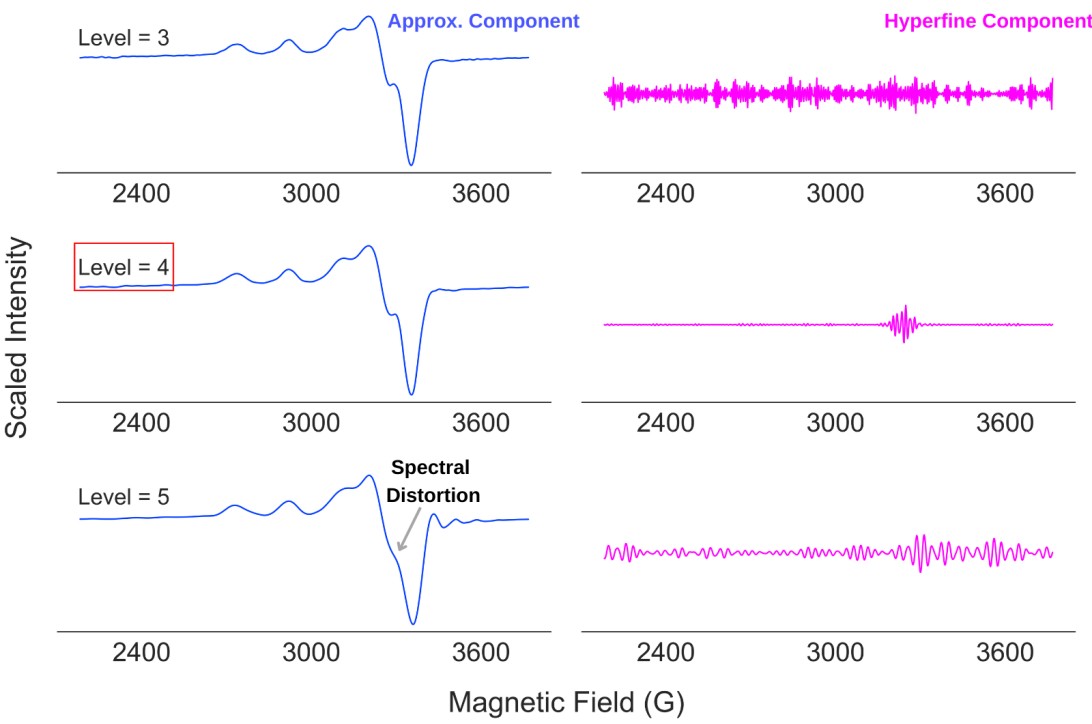

**Figure A1.** WPT analysis of ESR spectra of Cu-AHAHARA complex recorded at 100 K. The approximation and hyperfine components are shown at decomposition levels of 3, 4 and 5 using the Db9 wavelet. In this case, the approximation component at level-5 showed spectral distortion, implying that level-4 is the optimal decomposition level in this case.

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
