# Peer review of "A Simulation Independent Analysis of Single- and Multi-Component cw ESR Spectra"

_magnetochemistry, doi:10.3390/magnetochemistry9050112_

Round 1

Reviewer 1 Report

The authors present an interesting paper describing improved EPR spectral analysis based on wavelet packet-based algorithms. The paper is well structured, understandable and shows considerable work was put into testing the new algorithm in various real-life analysis scenarios. However, I would like the authors to address the following points:

1. A clear discussion of the limitations of the WPT method is missing, perhaps something like a table with direct comparison to spectral simulation, highlighting the pros and cons or capabilities of each method would be helpful. The reader might, for example, find out on line 276 that the WPT analysis might fail to reproduce relative intensities. Is this always the case, or only in case of overlapping spectra? Can the method be used to quantify the relative contribution of individual components in multi-component spectra? The results clearly show the excellent extraction of g-factor and hyperfine splittings in the presented compounds, but would the method work for systems with additional interactions, for example, zero-field splittings?

2. A somewhat related note, it is not clear to me how the authors distinguish between real components and noise. The authors claim that the method should be independent of the users apriori knowledge about the studied compound. In Figure 3 B1, we can see some additional small components between the first and second Tempo peaks. These are even more pronounced in Figure 4 A. Are these components real? When compared (in relative intensity) by a layman's eye to the hyperfine peaks in Figure 6 or Figure 7, they certainly seem comparable. In those systems, they are identified as real features, in Figure 3 and 4 they are not mentioned. What objective criteria could be used to distinguish between analysis artifacts and real features?

3. A detailed description of used software/algorithm/functions is missing. I presume Matlab was used, but which version? Is the WPT decomposition a built-in function or a custom one?

4. Finally, the authors conclude that they have presented a complete recipe for implementation of this method and hope for a wide adoption. I can also see the value of this method when used in concert with Easyspin simulations. As an average Easyspin user, not an algorithm developer, I would have trouble using this method as I wouldn't know where to even start. If the authors truly wish for wider adoption of their method, I suggest including an example script in the uploaded data or a supplementary document detailing the exact workflow.

Author Response

Response to Reviewer Comments

The authors present an interesting paper describing improved EPR spectral analysis based on wavelet packet-based algorithms. The paper is well structured, understandable and shows considerable work was put into testing the new algorithm in various real-life analysis scenarios. However, I would like the authors to address the following points:

Comment #1: A clear discussion of the limitations of the WPT method is missing, perhaps something like a table with direct comparison to spectral simulation, highlighting the pros and cons or capabilities of each method would be helpful. The reader might, for example, find out on line 276 that the WPT analysis might fail to reproduce relative intensities. Is this always the case, or only in case of overlapping spectra? Can the method be used to quantify the relative contribution of individual components in multi-component spectra? The results clearly show the excellent extraction of g-factor and hyperfine splittings in the presented compounds, but would the method work for systems with additional interactions, for example, zero-field splittings?

Author Response: We greatly appreciate this comment, as it is important for the comprehensive analysis of the ESR spectra using simulation-independent approach. It is worth mentioning that the present work is a part of a multi stage project. In our previous work, we showed that wavelet transforms can be used for hyperfine decoupling from poorly resolved spectra, which led to this work that focuses on ESR spectral analysis in terms of extraction of g-values and hyperfine coupling in a simulation-independent manner from partially resolved and unresolved spectra. The next steps in the project involves automated analysis, including the quantitative analysis. At this time, it can be carried out manually from the extracted wavelet components. Similarly, the relative intensities can be accurately determined for all cases; however, we focused on g and a values, a single wavelet component is sufficient to retrieve those values. For retrieving complete intensities, all the wavelet components (usually more than one) pertaining to a spectral feature are needed. In the attached figure, we have shown an experimental ESR spectrum (black) and the corresponding Approximation (red) and Detail (blue) components, obtained using stationary wavelet transform and Db9 wavelet at decomposition level = 4. The quantitative nature of the analysis is demonstrated by comparing the experimental and the reconstructed (Approx. + Detail) spectra (magenta). WPT spectral analysis involves 2^n number of components at a decomposition level of n, while in our algorithm, we picked only one of those components for analysis. This can be done manually, but automated analysis is preferable, which is the next step in the ongoing project.

For zero-field splitting, the WPT should be independent of it. In our approach, we are resolving the Detail components, where all the overlap is separated. The zero-field splitting effect can be observed in the Approximation component (low frequencies in the signal). This is an excellent point and we will add the spectra with zero-field splitting effect in the open-source big database we are creating. Although we wished, unfortunately we don’t have any such experimental spectra for the current work, but hope to have such spectra in future for more samples collected.

Pros and Cons: We indeed want to carry out the pros and cons studies, which we feel will be most useful once the fully automated approach is ready. It will allow us to compare a large spectral dataset with simulation methods such as EasySpin, NLLS and SpinFit. The comparison will be carried out against all the relevant measures, such as accuracy, user intervention, number components detected, SNR, complexity of spectra, among others. Crystal structure of some of the molecules would be used as a benchmark.

Comment #2: A somewhat related note, it is not clear to me how the authors distinguish between real components and noise. The authors claim that the method should be independent of the users a priori knowledge about the studied compound. In Figure 3 B1, we can see some additional small components between the first and second Tempo peaks. These are even more pronounced in Figure 4 A. Are these components real? When compared (in relative intensity) by a layman's eye to the hyperfine peaks in Figure 6 or Figure 7, they certainly seem comparable. In those systems, they are identified as real features, in Figure 3 and 4 they are not mentioned. What objective criteria could be used to distinguish between analysis artifacts and real features?

Author Response: Thanks for asking this question. The critical test used in this regard is calculation of the inter-peak spacings. For artifacts, the inter-peak spacing varied significantly. The process is now highlighted by adding the following sentences in the revised manuscript (line: 214 – 217):

“It should be noted that the hyperfine components contain some features in between the regions of interest, clearly visible in Figure 3 [B1]. Such artifacts can be discarded by either (i) comparing with the original spectra (which was the case for Figure 3 [B1]) or (ii) analyzing the inter-peak spacing, which yields non-uniform splitting in case of artifacts.”

In addition, Figure 7 of the original manuscript illustrates the basis of excluding some of the lines in the hyperfine component (Figure 7 caption):

“… The WPT component in the shaded region (gray) was not considered in the analysis because of varying inter-peak spacing in the region.”   

Comment #3: A detailed description of used software/algorithm/functions is missing. I presume Matlab was used, but which version? Is the WPT decomposition a built-in function or a custom one?

Author Response: Yes, indeed, MATLAB was used to carry out the analysis. WPT is a built-in function both in MATLAB and Python language. We updated the text with the following sentence in the revised manuscript (line: 126 – 127):

“The WPT analysis is performed using Matlab software, version 9.12.0.1884302 (R2022a) and an illustrative code is given in Appendix A.1.”

Also, an illustrative example with the Matlab code is added in the appendix of the revised version of the manuscript (page: 15-16).

Comment #4: Finally, the authors conclude that they have presented a complete recipe for implementation of this method and hope for a wide adoption. I can also see the value of this method when used in concert with Easyspin simulations. As an average Easyspin user, not an algorithm developer, I would have trouble using this method as I wouldn't know where to even start. If the authors truly wish for wider adoption of their method, I suggest including an example script in the uploaded data or a supplementary document detailing the exact workflow.

Author Response: Thanks for this useful suggestion. To address the reviewer’s concern, an illustrative example with the Matlab code is added in the appendix of the revised version of the manuscript (page: 15-16).

In near future, we plan to incorporate this method in the website, denoising.cornell.edu for improved and easy access. We are also creating a YouTube video to demonstrate how the process works.

Author Response

Reviewer Comment

The manuscript by Roy et al. describes a method for processing the EPR signal in continuous waves that allows the separation of some hyperfine and super hyperfine contributions. The method although published by the same authors in previous papers is here optimized for mixed species and low resolution EPR spectra.

The method is not well known in the community but seems to give very interesting results. The examples presented seem to confront theory and experiment.

The study is of sufficient interest to be published. However I am quite frustrated about the reproducibility of some of the results because I think there is a lack of information about the parameters used.

Indeed, when trying to reproduce the results of the manuscript using the parameters of the article, the result is not totally conclusive.

An example:

Cu-AHAHARA at 100K using level4 and db9 gives this:

The result with level3 is already closer but not yet conclusive:

The lack of information about the threshold is also a problem.

I suggest that the authors add all the reproducibility information to the numerical results.

In conclusion, I think the manuscript deserves to be published if the authors provide all the information to reproduce the numerical results.

Author Response: First, we want empathize with the reviewer, as we were also in the similar situation when we first used discrete wavelet transform for the simulation-independent analysis and which eventually led to the use of Wavelet Packet Transform. Looking at the figure, we realized the reviewer is using the NERD software with the Undecimated Discrete Wavelet Transform, instead of Wavelet Packet Transform. Before addressing the comment, we illustrate the difference between discrete wavelet transform, undecimated discrete wavelet transform (aka Stationary Wavelet Transform, SWT) and wavelet packet transform:

  1. Discrete Wavelet Transform: It provides lowest possible time(magnetic field)-frequency resolution by reducing the number of data point by 2 at each subsequent decomposition level. It yields one Approximation component and one Detail component at each decomposition level.
  2. Undecimated Discrete Wavelet Transform: It improves on the time (magnetic field) resolution by avoiding downsampling by 2; however, frequency resolution is still reduced at the subsequent decomposition levels. It yields one Approximation component and one Detail component at each decomposition level. One limitation is that the data length needs to be 2^N for using the Undecimated Discrete Wavelet Transform.
  3. Wavelet Packet Transform: It provides both high time (magnetic field) and frequency resolution without any downsampling. However, increased resolution come at the cost of increased number of wavelet components. It yields one Approximation component and 2^j -1 Detail components at j^th decomposition level. This is illustrated in Fig. 1B.

There are further improvements in resolution possible, which goes until Heisenberg’s time-frequency uncertainty box, but is currently not required for this study.

At present, the NERD software doesn’t support WPT (due to increased number of Detail components) and we are developing a separate software package of this method. We are also creating a YouTube video to illustrate the process.

Reproducing the Results: The easiest way to reproduce the results is to take WPT (wpdec) in Matlab software, and observe all the components which reveals the different spectral features. Figure 2 (the first figure attached at the end of this document) in the manuscript  illustrates the process workflow. The code to reproduce the example mentioned by the reviewer (Cu-AHAHARA at 100K using Db9) is now provided in the appendix (page 15–16). Another point to note is that, inverse wavelet transform of that component is required (while reducing other wavelet component to 0). This can be seen in the figure (the second figure attached at the end of this document), how SWT (or UDWT), SWT reconstructions and WPT reconstructions at level-3 and 4 yield different results, which led to reviewer confusion as well.

It must be noted that built-in code for DWT, UDWT and WPT yields Approximation and Detail components in different formats (due to their different applications), so it is advisable to read the Matlab documentation for extracting the relevant Approximation and Detail components. As expert users, we have experienced temporary migraines over it! The format may also change from Matlab to Python, including the built-in function name. NERD software package avoids this problem for DWT and UDWT, and our new software package on WPT will also be format independent.

Round 2

Reviewer 1 Report

Thank you for addressing my questions, I recommend the article for publication.